# Effect of Cervical Transcutaneous Spinal Cord Stimulation on Sensorimotor Cortical Activity during Upper-Limb Movements in Healthy Individuals

**DOI:** 10.3390/jcm11041043

**Published:** 2022-02-17

**Authors:** Ciarán McGeady, Monzurul Alam, Yong-Ping Zheng, Aleksandra Vučković

**Affiliations:** 1Centre for Rehabilitation Engineering, University of Glasgow, Glasgow G12 8QQ, UK; c.mcgeady.1@research.gla.ac.uk; 2Department of Biomedical Engineering, The Hong Kong Polytechnic University, Hung Hom, Kowloon, Hong Kong; yongping.zheng@polyu.edu.hk

**Keywords:** neuromodulation, transcutaneous spinal cord stimulation, electroencephalography, event-related desynchronisation, rehabilitation, posterior root muscle reflex

## Abstract

Transcutaneous spinal cord stimulation (tSCS) can improve upper-limb motor function after spinal cord injury. A number of studies have attempted to deduce the corticospinal mechanisms which are modulated following tSCS, with many relying on transcranial magnetic stimulation to provide measures of corticospinal excitability. Other metrics, such as cortical oscillations, may provide an alternative and complementary perspective on the physiological effect of tSCS. Hence, the present study recorded EEG from 30 healthy volunteers to investigate if and how cortical oscillatory dynamics are altered by 10 min of continuous cervical tSCS. Participants performed repetitive upper-limb movements and resting-state tasks while tSCS was delivered to the posterior side of the neck as EEG was recorded simultaneously. The intensity of tSCS was tailored to each participant based on their maximum tolerance (mean: 50 ± 20 mA). A control session was conducted without tSCS. Changes to sensorimotor cortical activity during movement were quantified in terms of event-related (de)synchronisation (ERD/ERS). Our analysis revealed that, on a group level, there was no consistency in terms of the direction of ERD modulation during tSCS, nor was there a dose-effect between tSCS and ERD/ERS. Resting-state oscillatory power was compared before and after tSCS but no statistically significant difference was found in terms of alpha peak frequency or alpha power. However, participants who received the highest stimulation intensities had significantly weakened ERD/ERS (10% ERS) compared to when tSCS was not applied (25% ERD; *p* = 0.016), suggestive of cortical inhibition. Overall, our results demonstrated that a single 10 min session of tSCS delivered to the cervical region of the spine was not sufficient to induce consistent changes in sensorimotor cortical activity among the entire cohort. However, under high intensities there may be an inhibitory effect at the cortical level. Future work should investigate, with a larger sample size, the effect of session duration and tSCS intensity on cortical oscillations.

## 1. Introduction

Transcutaneous spinal cord stimulation is a non-invasive neuromodulatory technique that has shown potential in reversing upper-limb paralysis in spinal cord injury (SCI) patients [1,2]. The technique often involves placing one or more cathode electrodes at and around the spinal level of injury to deliver high-frequency currents at sub-threshold intensities. It has been postulated that electrical interaction with a combination of structures, such as dorsal column fibres, the dorsal horn and posterior/ventral roots, decreases the motor threshold, making voluntary motor control easier through residual descending pathways [3,4,5]. When combined with conventional rehabilitative therapies such as physical practice, tSCS has led to lasting functional improvements [1,2,6]. The extent to which tSCS modulates corticospinal pathways, however, is still a matter of contention.

Numerous studies have investigated tSCS modulation at both the cortical and spinal level [7,8,9,10,11,12,13]. Benavides et al., for example, investigated cortical modulation by comparing motor evoked potentials (MEPs) induced by transcranial magnetic stimulation (TMS) before and after 20 min of tSCS. They found that MEP amplitudes tended to increase following stimulation, implying facilitation of the corticospinal tract. Ambiguities still exist surrounding tSCS-based neuromodulation, however. In a similar study, Sasaki et al. reported a null effect of tSCS on MEP amplitude, albeit with sessions of a shorter duration [10]. Both studies, and indeed the majority of similar studies, used MEP amplitudes to provide a metric of cortical excitability. Other measures, such as cortical oscillations, offer an alternative perspective on the physiological effects of tSCS. Although MEP and oscillation amplitudes have both been associated with motor cortical excitability, they are not strongly correlated, and likely reflect different neural processes [14,15]. Where cortical oscillations tend to reflect the induced excitability of large populations of cortical neurons, MEPs are affected by the global excitability of corticospinal pathways [14,16]. An understanding of how each measure is affected by tSCS will build a stronger foundation in which to guide future tSCS-based neurorehabilitation strategies. A further benefit of understanding the influence of tSCS on cortical oscillations concerns the use of brain–computer interfaces, which are increasingly being used in neurorehabilitation, often when combined with stimulation-based therapies [17,18]. Such BCI paradigms rely on distinct and consistent modulation of sensorimotor oscillations during imagined or attempted movement. Facilitated expression of sensorimotor oscillations may improve the performance of such systems [19,20].

As far as we are aware, no studies have yet considered tSCS-based neuromodulation in terms of sensorimotor cortical oscillations as measured from the electroencephalogram (EEG). Given reports of enhanced excitability of motoneuron and cortico-motoneuronal synapses through spinal stimulation, we would expect an expression of neuromodulation in terms of cortical oscillations, as is the case with other stimulation-based modalities such as functional electrical stimulation (FES) [21], and transcutaneous electrical nerves stimulation (TENS) [22]. The variety of modulation is a matter of conjecture, however. On the one hand, we may expect sensorimotor cortical excitation, as sensory afferent volleys may be amplified resulting in stronger activation of the somatosensory cortex. On the other hand, we may expect cortical inhibition given high-frequency spinal cord stimulation has been linked to suppression of nociceptive transmission [23]. At the very least, we would expect a quantifiable difference in sensorimotor cortical activity when tSCS is applied compared to when stimulation is not present. Therefore, the aim of the present study was to investigate if sensorimotor cortical activity during upper-limb movement could be modulated by short duration continuous tSCS.

To test this hypothesis, we had healthy volunteers perform upper-limb movements as continuous tSCS was delivered to the posterior region of the neck, using typical clinical stimulation parameters [1,2]. EEG was recorded simultaneously and sensorimotor dynamics were extracted in an offline analysis. The alpha frequency is the most dominant EEG feature during the resting state, and its event-related (de)synchronisation (ERD/ERS) has been associated with cortical activation during sensorimotor tasks, reflecting asynchronous neural firing [16,24,25]. We performed a side-by-side comparison of ERD/ERS with and without tSCS. A further hypothesis was that sensorimotor neuromodulation by tSCS would be subject to a dose effect where the modulation would be facilitated or attenuated as a function of time. We tested this by considering the ERD/ERS of alpha and beta frequency bands across movement repetitions. In addition to ERD during movement, we compared resting-state EEG before and after tSCS.

## 2. Materials and Methods

### 2.1. Participants

Thirty able-bodied volunteers (9 females, 21 males; 26.7 ± 3.0 years old) participated in this study. Exclusion criteria included musculoskeletal pathology of the upper limbs, metal or electronic implants, medications that influenced neural excitability (antiepileptic, antipsychotics, or antidepressants), allergy to the electrode material, epilepsy, and pregnancy.

Sessions were conducted at the same time of day to minimise baseline EEG variances and subjects were allowed to take breaks in between recording runs. Written informed consent was obtained from all participants. This study was approved by the Human Subjects Ethics Sub-committee of the Hong Kong Polytechnic University and conducted according to the principles and guidelines of the Declaration of Helsinki.

### 2.2. Experimental Protocol

Based on a two-day crossover design, participants underwent two sessions on different days. Both sessions had participants perform a 10 min upper-limb movement task as EEG and EMG were recorded from the sensorimotor region of the scalp and forearms respectively (see Figure 1A for an illustration of the experimental setup). Continuous tSCS was applied concurrently to the cervical region of the neck during only one of these sessions (Figure 2A). The order in which participants received both sessions was pseudo-randomised.

There were two parts to a session: (1) resting-state EEG recording, and (2) a movement execution task. Part (1) was performed before and after the movement task to investigate potential modulation of physiological markers. While recording, participants were required to sit still in an upright position, minimising all body and eye movements. Resting-state EEG was recorded for 90 s with eyes closed. The movement execution task was performed in an upright, seated position and had participants perform rhythmic right-hand, left-hand and bimanual finger flexion, as cued by an interface on a computer screen (Figure 1A). A rightwards-pointing arrow cued right-hand movement, a leftwards-pointing arrow cued left-hand movement, and a double arrow pointing both left and right cued bimanual movements. We included a bimanual condition as SCI patients often use both hands during tSCS training, and most activities of daily living include coordination of both hands [2,26,27]. Each movement was performed and sustained for four seconds and repeated 30 times, with a randomised 1.5 to 2.5 s inter-trial interval. The timing scheme is illustrated in Figure 2B. EMG was recorded from the forearm muscles to measure movement onset.

### 2.3. Electroencephalography (EEG)

Two g.USBamp biosignal amplifiers (g.tec, Schiedlberg, Austria) recorded EEG at 1200 Hz from 19 passive electrodes: Fz, FC3, FC1, FCz, FC2, FC4, C3, C1, Cz, C2, C4, CP3, CP1, CPz, CP2, CP4, Pz, POz, and Oz, according to the international 10–20 system (See Figure 1A,B) [28]. Electrode AFz was used as ground and the reference electrode was placed on the right earlobe. EEG was filtered with a band-pass (0.01–100 Hz) and a notch filter (50 Hz). Electrode impedances were kept below 5 kΩ throughout the recording session, and participants were instructed to minimise head and eye movements in order to ensure high fidelity recordings. Given the considerable artefacts produced by concurrent tSCS, conventional data-cleaning techniques were unsuitable [29]. For example, the high amplitude stimulation component meant that applying rejection thresholds on peak-to-peak amplitudes would eliminate segments of otherwise meaningful EEG. Hence, rejection thresholds were not used during pre-processing and instead strict adherence to the protocol outlined above was followed.

### 2.4. Electromyography (EMG)

To determine the onset of upper-limb movement, electromyography (EMG) was used to measure the activity of the extensor carpi radialis (ERC) muscles (See Figure 1D). Two electrodes (Ag/AgCl; F-301, Skintact, Innsbruck, Austria) were positioned on the dorsal side of each forearm, above the belly of the ERC, with a 20 mm inter-electrode distance. Ground electrodes were attached to the lateral epicondyles. EMG was recorded with the same biosignal amplifier outlined above (band-pass filter: 5–1200 Hz; notch filter: 50 Hz) to ensure synchronisation with EEG. Movement onset was defined as the moment EMG activity exceeded the mean of the resting phase plus two times its standard deviation for at least 100 ms [30].

### 2.5. Transcutaneous Spinal Cord Stimulation (tSCS)

Using a DS8R Biphasic Constant Current Stimulator (Digitimer, Hertfordshire, UK), spinal cord stimulation was delivered in bursts of ten 100 μs long biphasic rectangular pulses at a frequency of 30 Hz (see Figure 1B for an illustration of a single burst) [31]. A round 3.2 cm cathode electrode (Axelgaard Manufacturing Co., Fallbrook, CA, USA) was placed between the C5–C6 intervertebral space, placement reflective of upper-limb rehabilitation in clinical practice. Rectangular inter-connected anode electrodes (8.9 × 5.0 cm) were placed symmetrically on the shoulders, above the acromion (see Figure 1A for an illustration) [32]. We used feedback from the participant to determine the current intensity. Starting at 0 mA, the current was gradually increased in 2.5 mA increments until the participant verbally communicated their wish to stop increasing. Participants were asked before each incremental increase whether they would be able to tolerate the sensation for at least 30 s. If they were unable to tolerate the intensity, the current was reduced by one increment and was used for the remainder of the movement task. The area of discomfort varied across participants. Some participants reported that discomfort was focused under the cathode electrode; others found the contraction of back and neck muscles intolerable; some reported a combination of both. Across all participants, tSCS current intensity was on average 50 ± 20 mA, with a minimum and maximum current of 10 and 85 mA respectively.

### 2.6. Quantifying Sensorimotor Cortical Activity during tSCS

EEG was pre-processed offline with a 3rd-order Butterworth band-pass filter (1–40 Hz) and notch filter (50 Hz). Next, continuous EEG was segmented into epochs from −2 to 6 s relative to movement onset. The power spectral density across time and frequency was found using the multitaper method (1–25 Hz) with a resolution of 0.5 Hz. This analysis was performed with channels C3, C4, and the mean of C3 and C4, for right, left, and bimanual movements respectively. Time-frequency power was normalised with respect to a pre-movement baseline, defined as −1.25 to −0.25 s before movement and the average time-frequency powers were averaged across all subjects for each movement type. Statistical masking was added to time-frequency plots to display only power values which deviated significantly (*p* < 0.05) from baseline, as determined by a cluster-based permutation test.

We then separately considered the mean alpha (7–13 Hz) and beta (14–25 Hz) band ERD/ERS during two phases of movement: (1) movement initiation (0.5–1.5 s), and (2) sustained movement (1.5–3.0 s). We expected tSCS would strengthen ERD during the sustained movement phase, reflecting similar results observed using FES during motor imagery [19]. ERD values were averaged across movement phases and compared between stimulation conditions with a Wilcoxon signed-rank test, where *p* < 0.05 indicated a statistically significant difference in cortical activity.

A topographical analysis was performed by averaging the movement phases outlined above in the alpha and beta frequency bands for each recorded channel. The spatial distributions of cortical activation were used in a cluster-based permutation test to compare the ERD patterns while tSCS was on compared to when tSCS was off. A significance threshold of 0.05 was used to identify significant differences in topographical distributions between the two stimulation conditions.

Finally, in order to investigate a dose-effect of tSCS on cortical activity, we considered the correlation between ERD during each trial and sequence of trials by calculating Pearson’s correlation coefficient. A Wilcoxon signed-rank test was used to determine if the participant-wise average correlation coefficients significantly differed between stimulation conditions.

### 2.7. Neuromodulation of Resting-State EEG

We explored whether tSCS exerted a neuromodulatory effect on resting-state EEG by comparing individual alpha frequency before and after the movement task. We used resting state, eyes closed EEG and segmented it into one-second epochs with a 0.1 s overlap. Each epoch was windowed using a Hamming window and the periodograms ($2^{15}$ point FFT) were averaged to estimate the power spectral density (PSD). The alpha peak frequency was defined as the frequency with the maximum power in the 7–13 Hz range. The alpha peak frequency after the intervention was expressed as a percentage change from the alpha peak before the intervention. We also considered the power of the alpha peak and similarly normalised this with respect to pre-intervention power. A Wilcoxon signed-rank test was performed to determine if there was a significant difference in the change of alpha peak frequency and power between the tSCS-off and tSCS-on conditions.

## 3. Results

### 3.1. Event-Related (De)synchronisation (ERD/ERS)

To investigate the effect of tSCS on sensorimotor activity during movement we calculated alpha and beta band power differences with respect to rest. Figure 3 shows time-frequency power values averaged across all participants for left, right, and bimanual finger flexion. The plots only display power values that significantly differed (*p* < 0.05) from baseline, as determined by a cluster-based permutation test. Each movement type showed significant broadband (8–25 Hz) ERD with particular power suppression in the alpha band (8–12 Hz). Right and bimanual movements tend to show similar patters of ERD regardless of whether tSCS had been applied or not. Left-hand movements appeared to have deeper and more sustained alpha desynchronisation when tSCS was applied.

To test for a significant difference of ERD between conditions we divided each movement into two phases: (1) movement initiation (0.5–1.5 s after movement onset), and (2) sustained movement (1.5–3.0 s after movement onset). Figure 4 shows the average ERD during movement initiation for each movement type and stimulation condition in the alpha and beta bands. There were no significant differences detected in the alpha band (Figure 4A: Left: *p* = 0.15; Right: *p* = 0.14; Bimanual: *p* = 0.90), nor in the beta band (Figure 4B: Left: *p* = 0.77; Right: *p* = 0.60; Bimanual: *p* = 0.75). Although ERD shows variability, the variance is inline with other studies reporting ERD within participants and across sessions [33]. On average, however, there was a lack of consistency in the direction of modulation with some participants having stronger ERD with stimulation, and some having suppressed ERD.

Similar results are seen in Figure 5 which presents ERD values during sustained movement (Alpha: Left: *p* = 0.19; Right: *p* = 0.12; Bimanual: *p* = 0.40; Beta: Left: *p* = 0.4; Right: *p* = 0.90; Bimanual: *p* = 0.94).

### 3.2. Topographic Analysis of ERD

The ERD topographic patterns during movement initiation and sustained movement are illustrated in Figure 6 and Figure 7 respectively. It can be seen that there is desynchronisation present at all the electrodes in the alpha and beta frequency bands in both stimulation conditions. Figure 7A shows bilateral alpha ERD when tSCS is off. When tSCS is on the pattern appears more contralaterally dominant over C4 electrodes (Figure 7C). However, a cluster-based permutation test showed that there were no regions of the topographical distributions that significantly differed between conditions. This was the case for the beta band and for sustained movement shown in Figure 6.

### 3.3. Dose Effect of Event-Related Desynchronisation

We found that on average there was no dose effect of tSCS on alpha or beta ERD (Figure 8). Taken as a group, the average correlation coefficients were close to zero with or without the presence of tSCS. A Wilcoxon signed-rank test corrected for multiple comparisons found no significant difference between conditions in either frequency band (Alpha: *p* = 0.16; Beta: *p* = 0.75).

### 3.4. Resting State Modulation

We found that resting state individual alpha peak frequency was not significantly altered by tSCS (*p* = 0.67), showing an approximately 0% change from pre-intervention alpha for both stimulation conditions (Figure 9A). Furthermore, the change in alpha power was also unaffected by tSCS (*p* = 0.20), shown in Figure 9B.

### 3.5. Effect of tSCS Intensity

Given the variability across sessions shown in Figure 4 and Figure 5, we investigated whether the variance could partially be explained by stimulation current intensity, given intensity was tailored to the individual. Figure 10 shows that ERD/ERS appears similarly distributed between conditions at around 20% ERD for intensities between 10 and 60 mA. Intensities above around 65 mA, however, tended to result in suppressed ERD, or even ERS, relative to the tSCS-off condition. A linear regression found that ERD/ERS and tSCS intensity were indeed positively, and significantly, correlated (*r* = 0.409, *p* = 0.025).

The discomfort felt by participants tended to grow as a function of tSCS intensity. It may have been the case, therefore, that relative alpha power was being suppressed by the uncomfortable sensation, resulting in less desynchronisation during movement, a known consequence of pain on the alpha rhythm [34,35]. Suppression would likely have been more prominent in participants who received the highest intensities. To test this, we found the correlation between intensity and pre-movement relative alpha power (−1.5 s to −0.5 s relative to movement onset): *r* = −0.062, *p* = 0.75. Although the correlation was not significant, the participants who received the highest intensities tended to have reduced alpha power during rest.

Interestingly, when two sub-groups were formed from participants from the lower and upper 25% of the intensity distribution, ERD/ERS become significantly altered by tSCS in the high-intensity group only. Figure 11 shows that in the early phase of movement, ERD/ERS is significantly elevated, (*p* = 0.016) from around −25% without tSCS to around 10% during tSCS, reflective of (event-related) synchronisation rather than desynchronisation. This is seen also in the beta band (*p* = 0.015) and the trend is seen during sustained movement but without significance (*p* > 0.05).

Resting-state alpha frequency and power were also reevaluated in terms of current intensity but no altered effect was found.

### 3.6. Stimulation Adherence

Continuous tSCS was well tolerated by the majority of participants. In two cases, upon receiving tSCS at the beginning of the session, the sensation was considered overwhelming and the participants opted not to continue with the experiment. Both reported that, although not painful, stimulation was uncomfortable and made sitting still difficult.

## 4. Discussion

The present study showed that a 10 min session of tSCS did not significantly modulate sensorimotor brain rhythms during repetitive upper-limb movements. Similarly, resting-state EEG, as characterised by alpha-band peak frequency and power, was unaffected by continuous tSCS. An investigation of tSCS intensity revealed, however, that cortical activity may have been suppressed among participants who received the highest stimulation intensities, given ERD/ERS was significantly altered for these participants. This work suggests that tSCS intensity may be an important factor to elicit consistent modulation at the cortical level. However, as this high-intensity group is a subset of the overall participant sample, the sample number is small and must be verified on a larger scale.

The inter-participant and inter-session variability in measures such as ERD and alpha power, tended to reflect the inherent variances associated with these measures, as they are in line with other research [10,33]. However, the variance may partially be attributable to current intensity, which was individualised for each participant based on their maximum tolerance. This choice of protocol was based on typical clinical procedures for determining current intensity [2,36,37]. The alpha and beta ERD/ERS of participants who received the highest intensity stimulation tended to be weaker compared to when tSCS was not present. This may imply cortical inhibition following tSCS, which would echo similar claims made by Benavides et al. [7]. Conversely, this reduction in ERD/ERD may have been a consequence of the discomfort associated with high-intensity currents as reduced resting-state alpha power has been associated with exposure to painful sensations [34,35], and lower alpha often correlates with weaker ERD during movement [38]. It is difficult to speculate on the role tSCS intensity played on the individual as each participant received only one level of tSCS intensity. Future analyses should have each participant receive multiple current intensities in order to discern if an intensity-dependent effect exists.

Transcutaneous spinal cord stimulation must penetrate deep into spinal structures, passing multiple layers of skin, fat, muscle, and vertebrae, in order to exert a neuromodulatory effect [39]. Stimulation intensity must, therefore, be strong enough to overcome the impedance of the medium between electrodes. High-intensity stimulation, however, can result in intense discomfort or pain following the contraction of neck and back muscles, and activation of cutaneous pain receptors [40]. In this study, stimulation was set to the participants’ maximum tolerance. Maximum stimulation tolerance was shown by Manson et al. to constitute approximately 56% of the intensity required to induce a motor response [40]. This sub-threshold intensity is within the range that clinical studies have reported functional improvements following cervical tSCS [2,36,37]. It is possible, however, that participants with relatively poor stimulation tolerances did not receive activation of posterior-root afferents. This may explain the fact that neuromodulation of cortical oscillations was only observed in the subset of high-intensity participants. It may be the case, therefore, that tSCS, by its very nature, is unsuitable for a portion of a given sample. Future studies may need to consider exclusion criteria that eliminate participants who cannot tolerate stimulation intensities capable of spinal cord interaction.

Although this study is the first to investigate the effects of transcutaneous spinal cord stimulation on cortical oscillations, other studies have reported neuromodulation through electrical stimulation of peripheral musculature and nerves [21,41,42]. For instance, Insausti-Delgado et al. reported enhanced alpha and beta ERD during high intensity neuromuscular electrical stimulation of the wrist extensors [21]. They attributed this effect to the activation of muscle spindles and joint afferents which recruited proprioceptive fibres in the spinal cord, which in turn affected the motor cortex. Indeed, tSCS has been reported to also recruit large-to-medium proprioceptive fibres within posterior roots [39]. Yet the present study found that participants who underwent high-intensity tSCS displayed suppressed alpha and beta band ERD during movement, suggestive of inhibited cortical activity. It may be the case that high-frequency stimulation interfered with the conduction of sensory information to the somatosensory cortex, reducing cortical area activated during movement, which in turn resulted in decreased expression of alpha and beta ERD. Benavides et al. also noted cortical inhibition following cervical tSCS with a 5 kHz carrier frequency, and the effect was even more pronounced in SCI patients [7]. They attributed this inhibition to the activation of inhibitory cortical circuits which influenced motor cortical activity. It is unclear whether this inhibition is related to the reduction of cortical activity in the present study.

Some EEG-based investigations featuring electrical stimulation are challenging or impossible without applying artefact-attenuation techniques [43]. However, stimulation artefact contamination was not considered a confounding factor here as previous work by our group showed that, so long as the spectral region of interest does not overlap with the stimulation frequency, resulting EEG bares statistically similar characteristics to that of normal EEG [29]. Therefore, any differences found in spectral power would be attributable to endogenous neuromodulation and not signal corruption.

The lack of sham condition in this study may constitute a limitation given that the placebo effect has been shown to impact EEG-based metrics [44]. However, implementing a sham control with tSCS is non-trivial as the intensity range at which tSCS exerts a non-therapeutic effect is currently unknown. Similarly, non-therapeutic duration is also unknown, hence, protocols that ramp down after a brief period of stimulation were considered unsuitable. Further, the intense, non-painful sensation associated with tSCS, even at low currents, makes the ambiguity required for establishing an effective sham control difficult. Indeed, as Turner et al. showed using transcranial direct current stimulation, participants were aware of whether they were or were not receiving active stimulation throughout the experimental procedure [45]. We expect that placebo effect contamination to be low, however, as the procedure and equipment were identical in both sessions, and the outcome measures (ERD/ERS during movement) were not known by participants. Effective sham-blinding protocols should be verified in the future, perhaps by stimulating a spinal level that does not project to the motor pools under investigation.

A significant limitation of this study is that it lacks a clinical population. We note that studies that include a patient cohort in addition to healthy controls often reported more marked modulation in the SCI group [7]. It may be the case that, in healthy participants, a ceiling effect limits the recruitment of additional fibres as the cortical–spinal network is already being used to its fullest extent during movement. Additionally, an SCI cohort would allow for higher currents to be explored, owing to reduced sensitivity at and below the spinal level of injury. This would likely minimise the effect noted here whereby individuals receiving the highest intensities of tSCS exhibited reduced resting-state alpha power due to discomfort.

## 5. Conclusions

This study, for the first time, investigated cervical tSCS neuromodulation in terms of sensorimotor oscillations as measured by EEG. Our results showed that, on a group level, there was no consistent excitatory or inhibitory effect in terms of cortical activity during upper-limb movement. However, consistency appeared to emerge among participants who received the highest stimulation intensities. ERD, a measure of sensorimotor cortical activity, was diminished in these participants, potentially implying an inhibitory effect of tSCS at the cortical level. However, this sub-set of participants constitutes a small population size. Future work should, therefore, specifically investigate the effects of tSCS intensity on cortical oscillations. Additionally, future work should endeavour to determine the critical duration required for cervical tSCS to exert a measurable effect on sensorimotor cortical activity.

## Figures and Tables

**Figure 1 jcm-11-01043-f001:**
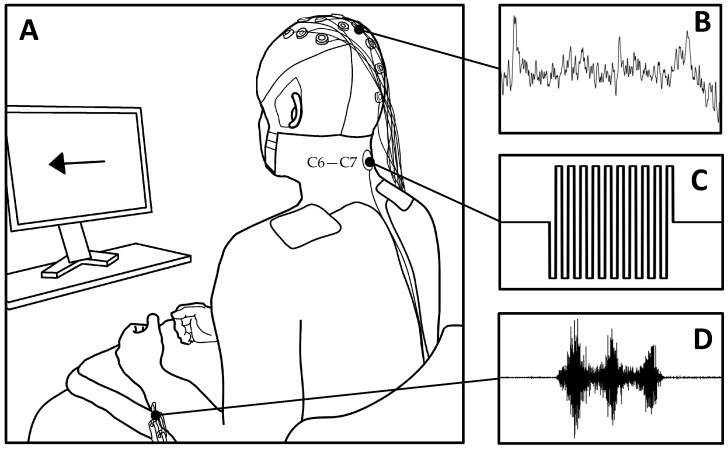
Experimental setup showing recording and stimulation modalities. (**A**) Participant receives cues from a computer screen to perform upper-limb movements. (**B**) EEG is recorded from the central area of the scalp. (**C**) One millisecond long burst containing 10 biphasic pulses is delivered at 30 Hz to the posterior region of the neck during continuous tSCS. (**D**) EMG during left-hand rhythmic finger flexion/extension over the extensor carpi radialis. The same setup was used on the right side.

**Figure 2 jcm-11-01043-f002:**
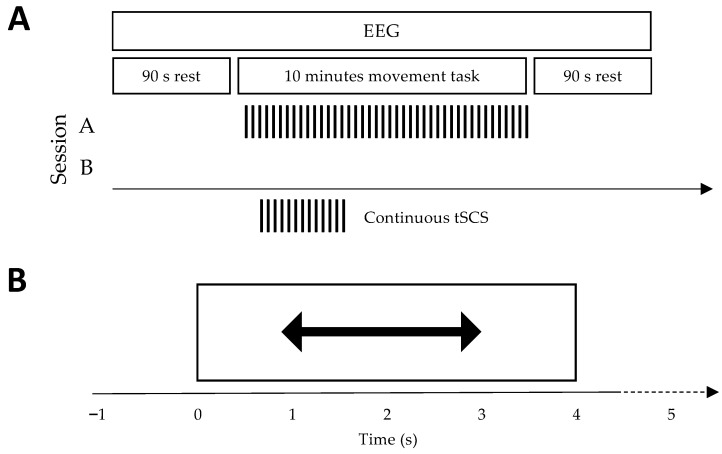
Experimental session protocol and movement task timing scheme. (**A**) Outline of the experimental sessions, carried out on different days. Both sessions began and ended with the recording of resting-state EEG with eyes closed. An upper-limb movement task lasted 10 min while EEG was recorded simultaneously. Only during session A was continuous tSCS applied to the cervical region of the spine. (**B**) The timing scheme of a single trial from the movement task. At t = 0 s an arrow appeared onscreen prompting the participant to perform either left, right, or bimanual finger flexion. The movement was sustained for four seconds. This was followed by a randomised 1.5–2.5 s inter-trial interval. There were 30 repetitions of each movement, totalling 90 trials.

**Figure 3 jcm-11-01043-f003:**
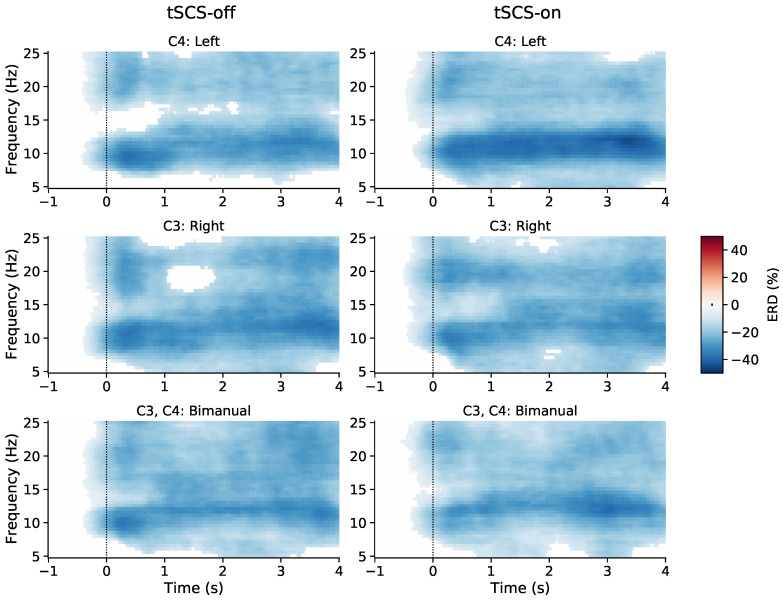
Time-frequency plots of event-related desynchronisation (ERD) during repetitive left, right, and bimanual finger flexion with and without tSCS. Only values significantly different from 0% ERD (*p* < 0.05) are shown, as determined by a cluster-based permutation test.

**Figure 4 jcm-11-01043-f004:**
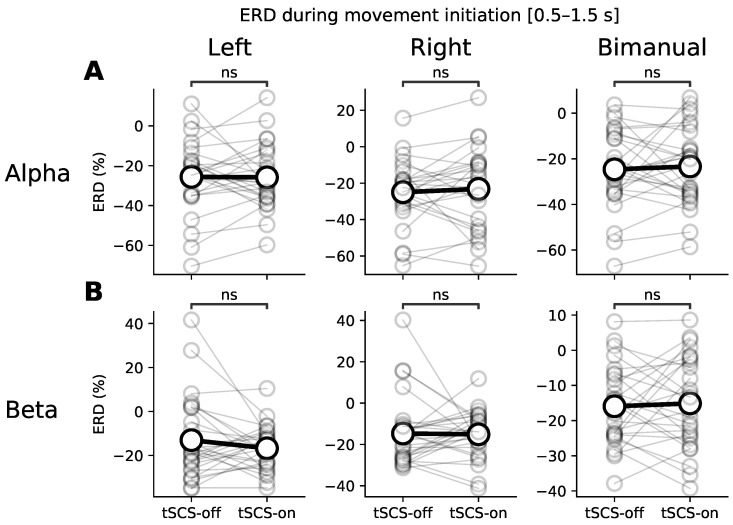
Average ERD during movement initiation (0.5–1.5 s) for each type of upper-limb movement (left, right, and bimanual finger flexion). (**A**,**B**) show ERD in the alpha and beta bands respectively. Grey markers show ERD of individual participants and the black markers show the participant-wise average. A Wilcoxon signed-rank test explored statistically significance differences between the tSCS-off and tSCS-on conditions for each movement and frequency band (‘ns’ denotes no significant difference).

**Figure 5 jcm-11-01043-f005:**
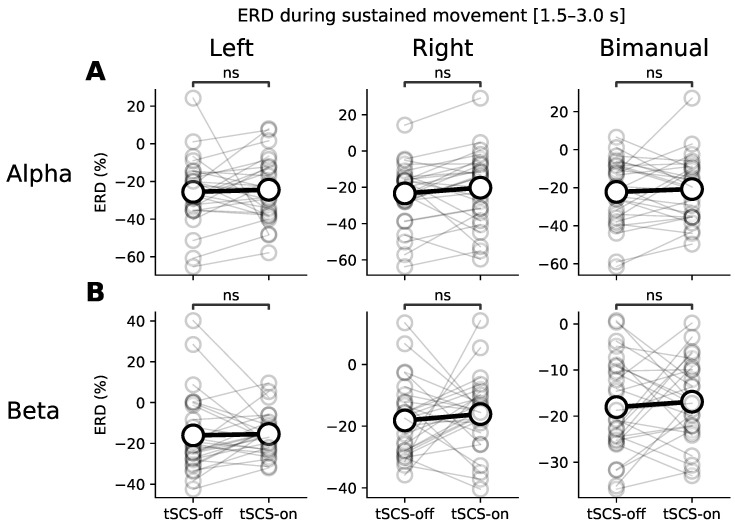
Average ERD during sustained movement (1.5–3.0 s) for each type of upper-limb movement (left, right, and bimanual finger flexion). (**A**,**B**) show ERD in the alpha and beta bands respectively. Grey markers show ERD of individual participants and the black markers show the participant-wise average. A Wilcoxon signed-rank test explored statistically significance differences between the tSCS-off and tSCS-on conditions for each movement and frequency band (‘ns’ denotes no significant difference).

**Figure 6 jcm-11-01043-f006:**
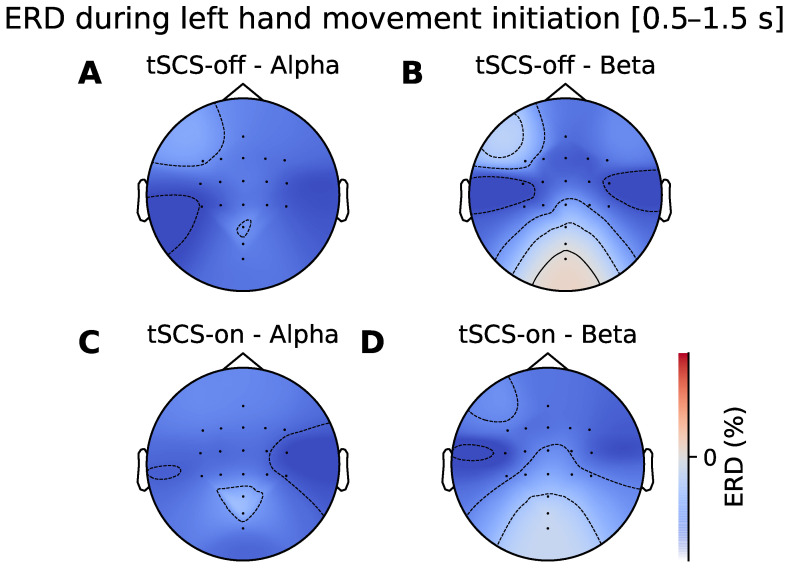
Topographic ERD/ERS distribution during left handed movement initiation (0.5–1.5 s after movement onset). (**A**,**B**) show spatial distribution of ERD/ERS in the alpha and beta bands without tSCS. (**C**,**D**) show the spatial distribution in the alpha and beta bands during with tSCS.

**Figure 7 jcm-11-01043-f007:**
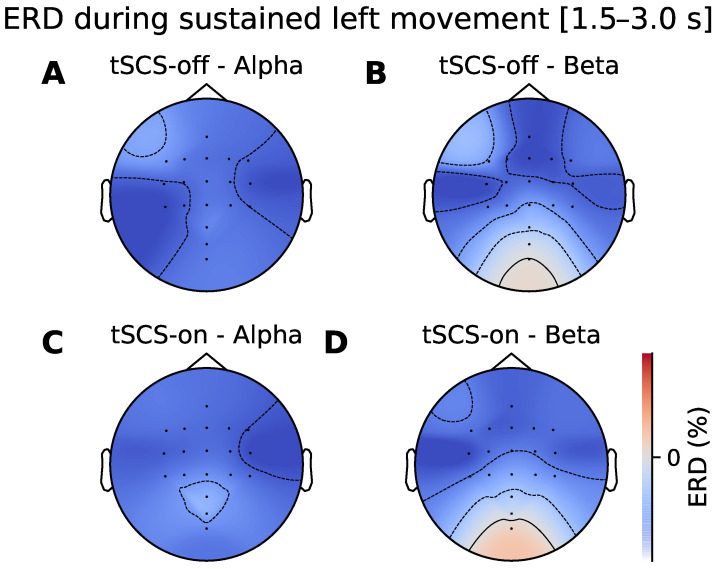
Topographic ERD/ERS distribution during sustained left handed movement (1.5–3.0 s after movement onset). (**A**,**B**) show spatial distribution of ERD/ERS in the alpha and beta bands without tSCS. (**C**,**D**) show the spatial distribution in the alpha and beta bands during tSCS.

**Figure 8 jcm-11-01043-f008:**
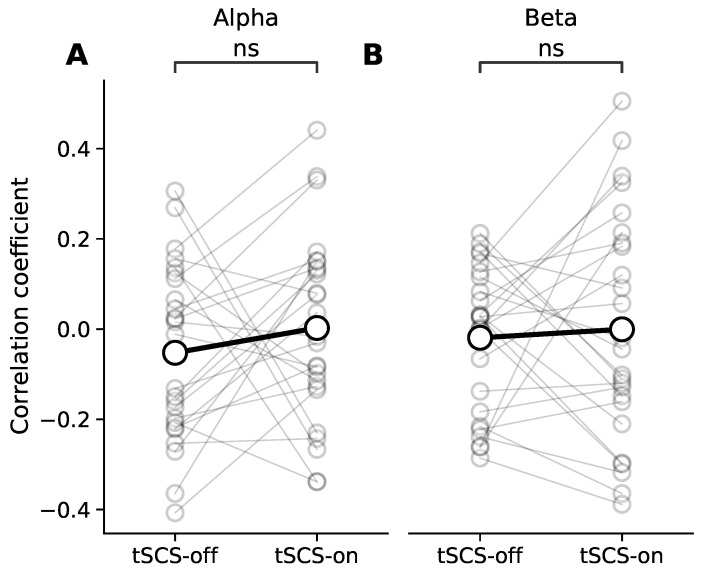
Correlation coefficient between event-related desynchronisation (ERD) in the alpha (**A**) and beta (**B**) bands during repetitive bimanual finger flexion and the sequence of trials in with and without tSCS. The grey markers represent correlation coefficient values for individual participants and the black markers represent the across-participant session average. ‘ns’ where there was no significant difference.

**Figure 9 jcm-11-01043-f009:**
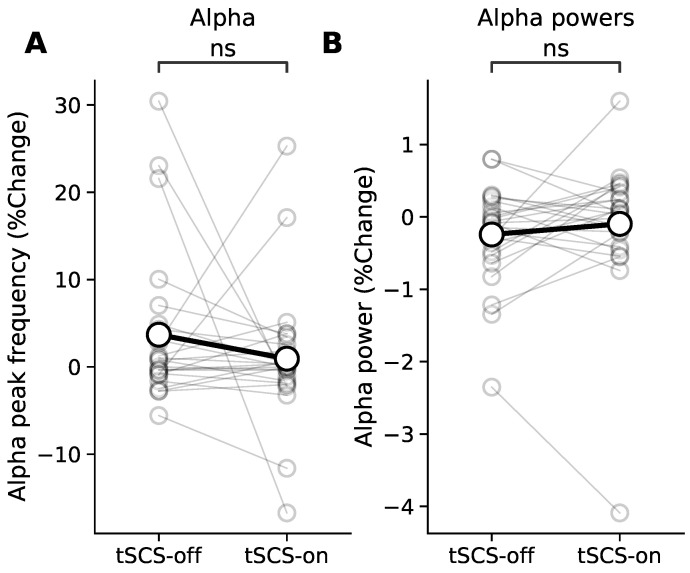
Resting state EEG. (**A**) shows the change (%) in peak alpha frequencies from baseline during resting state with eyes closed with and without tSCS. (**B**) shows the change in power of the peak frequencies. The grey markers represent individual participants and the black markers represent the across-participant session averages. Non-significance, as determined by a Wilcoxon signed-rank test, is expressed as ‘ns’.

**Figure 10 jcm-11-01043-f010:**
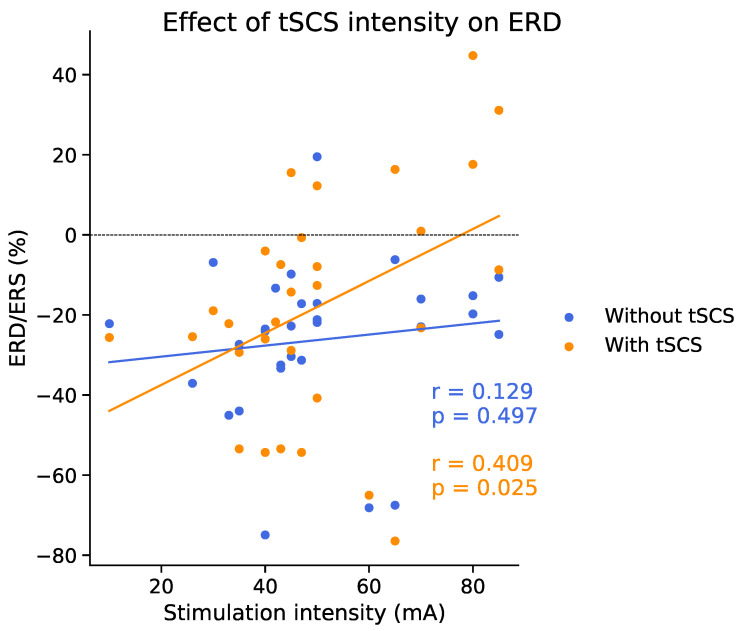
Subject-wise ERD/ERS (%) against tSCS intensity. To aid comparison, ERD/ERS values from the tSCS-off condition are also shown. Stimulation intensity relates to the tSCS-on condition only. Statistical outcomes from a linear regression are given in terms of *r* and *p* values.

**Figure 11 jcm-11-01043-f011:**
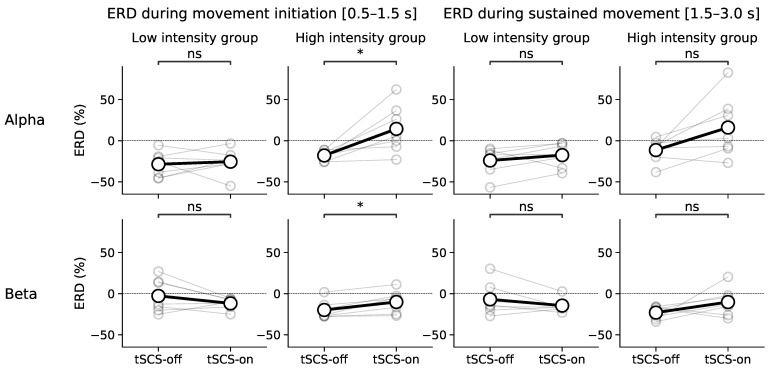
Event-related desynchronisation during movement with participants divided into two groups depending stimulation intensity. Low-intensity participants received tSCS at currents between 10 and 40 mA. High-intensity participants received tSCS at currents between 60 and 85 mA. A Wilcoxon signed-rank rest was used to determine statistically significant differences in ERD between experimental sessions. * *p* < 0.05, ns denotes non-significance.

## Data Availability

Data will be made available upon reasonable request to the authors.

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
