# Peer review of "Effect of Cervical Transcutaneous Spinal Cord Stimulation on Sensorimotor Cortical Activity during Upper-Limb Movements in Healthy Individuals"

_jcm, 2022, doi:10.3390/jcm11041043_

Round 1
Reviewer 1 Report
In the reviewed article, McGeady et al. aimed at investigating the neuromodulation of corticospinal pathways by transcutaneous spinal cord stimulation (tSCS) applied at the cervical levels. Changes in the sensorimotor cortical rhythms measured from the electroencephalogram before and after tSCS applications were used as a marker of this neuromodulation. The results show that the tSCS actions were non-consistent between tested subjects and on average did not evoke changes in the EEG recording.
The article is quite interesting and well written. However, I have a few concerns:
Major.
Introduction:
The rationale behind undertaking this study is not clear. In the introduction, the authors clearly state that the neuromodulatory effects of tSCS were already found by other groups on both cortical and spinal levels. However, despite this, they “expect an expression of neuromodulation at the level of the brain”. If cortical neuromodulation is already confirmed, then what is the rationale for this study?
What kind of added value will this study bring to neuroscience? If the novelty is to introduce the EEG measurement as a method for evaluating the tSCS effects, then what is the advantage of this technique in comparison to the ones already used such as measurement of MEPs elicited by TMS? The simple fact that EEG measurements were not previously used is not strong enough to justify undertaking this study
The authors state that “as far as we are aware, no studies have yet considered neuromodulation in terms of sensorimotor cortical oscillations as measured from the electroencephalogram (EEG)”. This is not exactly correct as Niso et al. 2021 (doi: 10.3389/fneur.2021.694310) have used the EEG recordings to test the impact of tSCS on somatosensory evoked potentials. Again, what is the rationale for undertaking this research?
Methods:
The methods are clearly written and give a lot of details on how the study was executed. However, I did not find any information if the participants were blind to the experimental conditions. There is no indication of blinding, nor there is any information of sham control. The authors write that the sessions were pseudo-randomized for each subject, but was the subject aware if the tSCS was applied or not?
Importantly, the “Stimulation adherence” paragraph, clearly indicates that the participants were aware of the experimental conditions. This is a major limitation of this study. If the participants were aware of the experimental conditions then a significant placebo effect might have occurred. The placebo effects can not be treated lightly as their effects on the EEG parameters are well documented (Petersen and Puthusserypady, 2019, 10.1109/EMBC.2019.8857549; Li et al. 2016, https://doi.org/10.3389/fncom.2016.00045). As such, the obtained EEG measurements may be contaminated by the placebo effect and as such unreliable.
Results:
With regard to the above, it is difficult to judge the results. However, the fact that the direction of EEG modulation seems to be extremely variable between participants points to some other factors (placebo) affecting the measurements.
Discussion:
The authors do not comment on the strong variability of the presented data. The majority of the discussion focuses on the average lack of effects of the tSCS on the whole studied population. Can the authors comment why some participants show strong modifications of the EEG signal, while others do not? Are there any additional between-patients variables that can explain these differences? Also, have the authors considered using a different polarity of the active electrode?
Reviewer 2 Report
In the summary, it would be necessary to include numerical statistical data in the part where the results are explained.In the first paragraph of section 2.3, I would recommend including some bibliographic reference confirming that the instrument used for the measurement is used in other studies or similar measurements.
The same occurs with the following sections where it is explained how the intervention is carried out (sections 2.4 and 2.5).
Was descriptive data collected from the participants? Some of them should be mentioned in the results.
I consider that the discussion is poor in bibliographic references. The data can be further contrasted with more similar studies. The final number of references is low.
Round 2
Reviewer 1 Report
The authors are to be congratulated for their hard work in improving the manuscript.
The majority of my major concerns were answered. The rationalization for conducting this study is now clearly stated in the introduction. Although the placebo problem still persists, the Authors now acknowledge this study limitation and discuss the possible consequences.
Of particular notice is the additional analysis of the intensity-dependent effects of tSCS. The re-analyzed results are now a lot more consistent and convincing.
One minor comment still remains: Line 34 “ It has been postulated that electrical interaction with dorsal root motorneurons” This sentence implies that motoneurons are located in the dorsal root. I’m positive that the authors are aware that spinal motoneurons are located in the ventral horns of the spinal cord grey matter (in the Rexed lamina IX), and in the motor nuclei of the brain-stem.
Please rephrase or correct this sentence.
Author Response
We thank Reviewer 1 for bringing this inaccuracy to our attention and have ammended the sentence in question. Line 34 now reads:
It has been postulated that electrical interaction with a combination of structures, such as dorsal column fibers, the dorsal horn and posterior/ventral roots, elevates the motor threshold, making voluntary motor control easier through residual descending pathways.
This manuscript is a resubmission of an earlier submission. The following is a list of the peer review reports and author responses from that submission.